# A Pilot Study of Multiplex Ligation-Dependent Probe Amplification Evaluation of Copy Number Variations in Romanian Children with Congenital Heart Defects

**DOI:** 10.3390/genes15020207

**Published:** 2024-02-05

**Authors:** Alexandru Cristian Bolunduț, Florina Nazarie, Cecilia Lazea, Crina Șufană, Diana Miclea, Călin Lazăr, Carmen Mihaela Mihu

**Affiliations:** 11st Department of Pediatrics, “Iuliu Hațieganu” University of Medicine and Pharmacy, 400370 Cluj-Napoca, Romania; 2Department of Medical Genetics, “Iuliu Hațieganu” University of Medicine and Pharmacy, 400012 Cluj-Napoca, Romania; 31st Pediatrics Clinic, Emergency Pediatric Clinical Hospital, 400370 Cluj-Napoca, Romania; 4Medical Genetics Compartment, Emergency Pediatric Clinical Hospital, 400370 Cluj-Napoca, Romania; 5Department of Histology, “Iuliu Hațieganu” University of Medicine and Pharmacy, 400012 Cluj-Napoca, Romania

**Keywords:** congenital heart defects, copy number variations, multiplex ligation-dependent probe amplification, 22q11.2 deletion

## Abstract

Congenital heart defects (CHDs) have had an increasing prevalence over the last decades, being one of the most common congenital defects. Their etiopathogenesis is multifactorial in origin. About 10–15% of all CHD can be attributed to copy number variations (CNVs), a type of submicroscopic structural genetic alterations. The aim of this study was to evaluate the involvement of CNVs in the development of congenital heart defects. We performed a cohort study investigating the presence of CNVs in the 22q11.2 region and *GATA4*, *TBX5*, *NKX2-5*, *BMP4*, and *CRELD1* genes in patients with syndromic and isolated CHDs. A total of 56 patients were included in the study, half of them (28 subjects) being classified as syndromic. The most common heart defect in our study population was ventricular septal defect (VSD) at 39.28%. There were no statistically significant differences between the two groups in terms of CHD-type distribution, demographical, and clinical features, with the exceptions of birth length, weight, and length at the time of blood sampling, that were significantly lower in the syndromic group. Through multiplex ligation-dependent probe amplification (MLPA) analysis, we found two heterozygous deletions in the 22q11.2 region, both in patients from the syndromic group. No CNVs involving *GATA4*, *NKX2-5*, *TBX5*, *BMP4*, and *CRELD1* genes were identified in our study. We conclude that the MLPA assay may be used as a first genetic test in patients with syndromic CHD and that the 22q11.2 region may be included in the panels used for screening these patients.

## 1. Introduction

Congenital heart defects (CHDs) are one of the most common congenital defects, with a global prevalence of 9.41 per 1000 births, increasing by approximately 10% in the last 15 years [1]. They are a complex group of disorders, arising from errors in the normal embryologic development of the cardiovascular system. The etiopathogenesis of CHDs is mostly still under intense research, but a multifactorial origin is widely accepted, with known genetic factors being responsible for around one-third of all cases [2]. They are classified as syndromic if there is an extracardiac major congenital defect, developmental delay, or facial dysmorphism associated and isolated if none of the characteristics mentioned above are present [3,4].

Copy number variations (CNVs) are submicroscopic structural genetic alterations defined as variations in the number of copies of DNA fragments larger than 1000 base pairs (bps) compared to the reference genome, including deletions, duplications, and other complex gains or losses of genetic material [5]. About 10–15% of all CHDs can be attributed to CNVs, the most common of them being 22q11.2 deletion, 7q11.23 deletion, 5p15.2 deletion, 1q21.1 deletion/duplication, and others [2,6]. Multiple studies have shown a higher burden of these submicroscopic structural genetic abnormalities in syndromic patients, ranging between 5 and 20% [6,7,8,9], compared to isolated CHDs, where the reported frequencies are between 3 and 12% [6,7,9,10,11,12].

The development of the cardiovascular system is orchestrated by a great number of genes, coding for different transcription factors and morphogenetic proteins that interact with one another in a complex and still poorly understood way [13]. The transcription factors *GATA4*, *TBX5*, and *NKX2-5* are key regulators of cardiomyocyte differentiation [14,15,16]. Point mutations in these genes were associated with different types of CHDs, specifically atrial and ventricular septal defects for *TBX5*; atrial and ventricular septal defects, pulmonary stenosis, or pulmonary atresia and atrioventricular septal defects for *GATA4*; and outflow tract defects (double-outlet right ventricle, transposition of great arteries, and tetralogy of Fallot) and atrial and ventricular septal defects for *NKX2-5*, respectively [17]. Signaling molecules encoded by genes like *BMP4* and *CRELD1* also act as regulators of different stages in heart development. A link between single nucleotide polymorphisms in the *BMP4* gene and septal defects, both atrial and ventricular, was demonstrated in a Han Chinese population [18]. Variations in the *CRELD1* gene were associated with atrioventricular septal defects, particularly in Down syndrome patients [19]. There is limited data regarding the involvement of CNVs in all these genes in the pathogenesis of CHDs.

The aim of this study was to evaluate the involvement of CNVs in the development of congenital heart defects by investigating the presence of CNVs in the 22q11.2 region and the *GATA4*, *TBX5*, *NKX2-5*, *BMP4*, and *CRELD1* genes in patients with syndromic and isolated CHDs and evaluating the association between genotype and phenotype in this population.

## 2. Materials and Methods

### 2.1. Patients

The study cohort was represented by children under 18 years old diagnosed with a CHD randomly selected over a 12-month period from the 1st Pediatrics Clinic, Emergency Pediatric Clinical Hospital, Cluj-Napoca, Romania.

For each subject, data were collected through clinical and echocardiographic evaluation. Transthoracic echocardiography was performed on all patients by two pediatric cardiologists in standard fashion using a commercially available system (Vivid S7, General Electric, Wauwatosa, WI, USA). All cardiac measurements were conducted in accordance with the recommendations of the European Association of Cardiovascular Imaging.

Based on the evaluation, the patients were divided into two study groups. In the first group, named syndromic, were included subjects with a confirmed diagnosis of a CHD and at least one of the following: extracardiac congenital defects, dysmorphism, or developmental delay. Patients with a confirmed genetic syndrome known to be associated with a CHD (e.g., Down syndrome, Turner syndrome) were excluded from the study. The subjects with a confirmed diagnosis of CHD but without any of the characteristics mentioned above were included in the second study group, named isolated.

Signed written informed consent was obtained from the legal guardian for all the subjects included. The study was approved by the Ethics Committee of “Iuliu Hațieganu” University of Medicine and Pharmacy, Cluj-Napoca (AVZ108/2022). The study followed the ethical guidelines of the Declaration of Helsinki.

### 2.2. DNA Extraction and MLPA Analysis

A peripheral venous blood sample was taken from all the patients, and DNA extraction was performed using the Wizard Genomic DNA Purification Kit (Promega, Madison, WI, USA), according to the manufacturer’s instructions. The quantity and quality of the isolated DNA were determined using a NanoDrop UV–Vis spectrophotometer (Thermo Fisher Scientific, Waltham, MA, USA).

Multiplex ligation-dependent probe amplification (MLPA) was performed using the SALSA MLPA P311-B2 Congenital Heart Disease probemix (MRC-Holland, Amsterdam, the Netherlands), according to the manufacturer’s recommendations, on an Applied Biosystems 3500 Genetic Analyzer (Thermo Fisher Scientific, Waltham, MA, USA). The P311-B2 CHD probemix contains 41 probes used for detecting CNVs in the following genes and regions: the 8p23 region—8 probes for the *GATA4* gene (1 probe for each of exons 1, 3, 4, 5, 6, and 7 and 2 probes for regions upstream of exon 1) and 1 probe for the *CTSB* gene, located downstream; the *NKX2-5* gene (5q35)—4 probes (2 probes for each of the two exons of the gene); the *TBX5* gene (12q24)—10 probes (1 probe for each of exons 1, 2, 3, 5, and 6 and 2 probes for each of exons 8 and 9 and 1 probe for intron 1); the *BMP4* gene (14q22)—4 probes (1 probe for each of exons 1, 3, and 4 and 1 probe for intron 1); the *CRELD1* gene (3p25)—2 probes (exons 4 and 11); the 22q11.2 region—3 probes (the *CDC45*, *GP1BB*, and *DGCR8* genes); and 9 reference probes. The sequences of the probes used are provided in the Appendix A. The data were analyzed using Coffalyser software v.220513.1739 (MRC-Holland, Amsterdam, The Netherlands), and the results were interpreted based on the dosage quotient (DQ), as loss of genetic material for a DQ < 0.65 and gain of genetic material for a DQ > 1.3, according to the product description.

Statistical analysis was performed using R-3.1.1 and GraphPad Prism 8 software. Nonparametric (Fisher’s exact and Mann–Whitney *U*) tests were performed after normality was assessed using the Shapiro–Wilk test, with a *p*-value limit of 0.05.

## 3. Results

A total of 56 patients were included in the study, with a median age of 35 months old (Q1–Q3: 5.75–111 months) and an approximately equal distribution among the sexes, as 32 patients (57.14%) were females. A family history of CHD was present in a total of 15 patients (26.78%), and 1 subject was born from consanguineous parents. The most common CHD in our study population was the ventricular septal defect (22 subjects—39.28%), followed by the tetralogy of Fallot and pulmonary stenosis (both with 7 subjects—12.5% each). Two patients had associations of more than two other types of CHDs, termed complex CHD (Figure 1).

Half of the subjects (28 patients) were classified as syndromic based on the criteria mentioned above. Dysmorphic features were the most common characteristic of the syndromic patients (21/28 subjects), including combinations of the following: low nasal bridge (8/28), low-set ears (7/28), micro-retrognathia (6/28), up-slanted palpebral fissures (6/28), hypertelorism, down-slanted oral commissures, and a high-arched palate. Extracardiac congenital defects were present in 12/28 subjects, the most common being bone deformities (syndactyly, radius dysplasia, and sternochondral dehiscence—4/28) and genitourinary malformations (hypospadias, posterior urethral valve, and cryptorchidy—3/28). Other congenital defects, namely vein of Galen malformation, anal atresia, omphalocele, laryngomalacia, and branchial cleft cyst, were each present in one subject. Situs inversus was associated in one of the patients. Developmental delay was present in 11/28 of the syndromic subjects.

There was no statistically significant difference between the syndromic and isolated groups based on the clinical characteristics of the patients, with the except of birth length and length and weight at the time of blood sampling. No difference was observed in the distribution of CHDs between the two groups (Table 1).

The MLPA analysis revealed the presence of CNVs in two of our subjects (3.57%), both from the syndromic group (Figure 2). Case no. 1 was a 5-year-old girl (at the time of blood sampling) with interrupted aortic arch type B and a perimembranous ventricular septal defect (VSD), surgically corrected at the age of 5 days, associated with craniofacial dysmorphism, characterized by small and low-set ears, low nasal bridge, up-slanted palpebral fissures, macroglossia, long philtrum, and mild developmental delay, especially regarding language acquisition. The MLPA analysis showed a heterozygous deletion of the *CDC45*, *GP1BB*, and *DGCR8* genes, representing a DNA fragment of the minimum 630 kbp from the 22q11.2 region.

Case no. 2 was a 3-year-old boy (at the time of blood sampling) with a perimembranous VSD; a mild facial dysmorphism represented by low-set ears, low nasal bridge, short palpebral fissures, and down-slanted oral commissures; and no extracardiac congenital defects. He also presented an important somatic growth delay, with a Z-score of −3.4 for height to age and −5.98 for weight to age (at the time of blood sampling). The same CNV, represented by a heterozygous deletion in the 22q11.2 region, was detected by the MLPA analysis.

No CNVs involving the *GATA4*, *NKX2-5*, *TBX5*, *BMP4*, and *CRELD1* genes were identified in our study. Also, there were no CNVs detected in any of the investigated regions in the isolated group.

## 4. Discussion

Copy number variations have been in the attention of researchers as an important source of genetic variation that determines an increased susceptibility for developing a CHD, since there is a high burden of known and rare CNVs in these patients (including de novo ones) compared to the healthy controls [20]. In this study, we investigated the presence of CNVs in the 22q11.2 region and the *GATA4*, *TBX5*, *NKX2-5*, *BMP4*, and *CRELD1* genes in patients with CHDs and discovered two cases with heterozygous deletion of the 22q11.2 region.

The involvement of 22q11.2 deletion in the development of heart defects is well studied [21,22]. It is one of the most common CNVs, with an overall prevalence of approximately 1:6000 births [22] because of the high mutational rate of the locus due to its particular genomic architecture, with low copy repeats flanking the region and predisposing to non-allelic homologous recombination, a mechanism known to generate CNVs [20,21,23]. Up to 80% of the patients carrying the 22q11.2 deletion have congenital heart defects, the most common being conotruncal heart defects, such as tetralogy of Fallot, interrupted aortic arch type B, perimembranous ventricular septal defect, and truncus arteriosus [22,24]. On the other hand, 11.6% of patients with conotruncal heart defects carry the 22q11.2 deletion, with the highest association in interrupted aortic arch type B (approximately 50%) [20,24]. Our findings are concordant with the published data, both patients carrying the CNV being diagnosed with perimembranous VSD and one of them associating interrupted aortic arch type B.

There are a great number of genes and regulatory elements in the 22q11.2 region. The MLPA kit used in the present study employed primers for three of those genes (*CDC45*, *GP1BB*, and *DGCR8*), covering approximately 630 kbp. Figure 3 describes the genetic architecture of the deleted DNA fragment found in the two patients presented. This fragment also includes the *TBX1* gene, which haploinsufficiency is known to be one of the most important determinants of the embryologic changes characteristic in 22q11.2 deletion syndrome, including heart defects [25]. Moreover, the *DGCR8* gene has been linked to the development of CHDs due to its reduced expression in these patients, independent of the presence of 22q11.2 deletion syndrome [26,27]. *DGCR8* is a key component of miRNA biogenesis, facilitating miRNA maturation [28], and thus, the altered expression determines a dysregulation in a number of miRNAs (miR-1a, miR-133a, miR-134, miR-143, miR-145a, and miR-194) involved in heart development [26,29].

Studies have proposed the classification of CHDs as syndromic based on the association of at least one of the following criteria: the presence of an extracardiac congenital defect, developmental delay, or facial dysmorphism [3,4]. Our data showed that the Z-scores for the birth length and also for the height and weight at the time of blood sampling were lower in the syndromic group compared to the isolated cases, the difference being statistically significant. Although no significant association was made between short stature, defined as a Z-score for heights lower than −2, and syndromic CHD, probably due to the small number of patients in our study cohort, there seems to be a link with abnormal stature development. More research is needed to evaluate the opportunity of including abnormal growth patterns as a criterion of defining syndromic CHDs.

The CNV burden is generally higher in syndromic CHDs compared to isolated cases, with multiple studies reporting various frequencies, ranging between 5 and 20% for syndromic [6,7,8,9] and 3 and 12% for isolated CHDs [6,7,9,10,11,12]. In our study, both of the patients carrying the CNVs formed the syndromic group, with a reported CNV proportion (2/28) comparable with the data published in the literature for this specific group. Breckpott et al. raised awareness that the percentage of CNVs in the isolated CHD population may be overestimated due to a retrospective diagnosis of syndromic CHD after CNV identification [7]. Because some of the features that characterize syndromic patients may appear later in life or may be very attenuated (especially dysmorphic traits) [7,20], attention must be given to classify as accurately as possible every patient for more personalized care.

Identification of the genetic background of CHDs has great clinical importance, because it allows for an early specific diagnosis and a more comprehensive care for patients with syndromic CHDs, and it helps in the assessment of recurrence risk and helps evaluate the prognosis and the outcomes of different therapeutic measures for the patient [31,32]. For example, patients with 22q11.2 deletion have affected survival and complication rates in the case of CHD repair, requiring longer cardiopulmonary bypass times and longer postoperative stay in the intensive care unit, with globally worse surgical outcomes [32]. The presence of potentially pathogenic CNVs is equally important in isolated cases of CHDs, their association increasing the risk of death and heart transplant by 2.6 times after heart surgery [12].

Multiplex ligation-dependent probe amplification (MLPA) is a polymerase chain reaction (PCR)-based genetic testing method designed to determine the copy number of up to 40–50 different DNA sequences in one reaction [33]. Monteiro et al. proposed using MLPA as the first test for the genetic screening of patients with syndromic CHD due to its high efficiency in detecting chromosomal abnormalities, excluding aneuploidies (11/12 of the pathogenic and probably pathogenic CNVs in their study), lower costs, and accessibility in terms of performing the test and analyzing the results [34]. Nagy et al. confirmed the previous findings by providing a diagnostic pathway for patients with syndromic CHDs in which MLPA plays an important role ahead of more expensive techniques such as a microarray after a meticulous phenotyping of the patient by a specialized clinical geneticist [31].

Table 2 summarizes the current knowledge regarding MLPA evaluation of CNVs in our regions of interest (the 22q11.2 region and *GATA4*, *TBX5*, *NKX2-5*, *BMP4*, and *CRELD1* genes) in different populations of CHD patients. Most of the studies that reported the detection of CNVs identified variations in the 22q11.2 region [25,29,30,31,32,33], both deletions and duplications, demonstrating once again the high mutagenicity of this genetic locus, as described previously [20,21].

There is a complex network of morphogens and transcription factors that orchestrates the development of the cardiovascular system, in which *GATA4*, *NXK2-5*, and *TBX5* act as early regulators [13]. In vitro studies on cell cultures have shown that the combination of these three transcription factors is sufficient for activating the genetic program to induce cardiac specification [15,42].

*GATA4* is a member of the highly conserved GATA family, characterized by the presence of two zinc-finger DNA-binding domains [43]. It plays an important role in the development of the atrioventricular region, including valvular development, due to its high expression at the level of endocardial cushions [44,45]. Genetic alterations of *GATA4*, especially single nucleotide polymorphisms (SNPs), have been linked to CHDs in the human population, associations being made with atrial septal defects (ASDs), ventricular septal defects, atrioventricular septal defects, tetralogy of Fallot, pulmonary stenosis, and pulmonary atresia [46,47,48,49,50]. Studies have also reported CNVs involving this gene. Floriani et al. described the presence of a deletion involving all of the seven exons of *GATA4* in two patients with complex CHDs (persistent left superior vena cava, ostium secundum ASD, perimembranous VSD, and abnormality in the pulmonary artery—pulmonary stenosis or bicuspid pulmonary valve, respectively), both associated with extracardiac findings (craniofacial dysmorphism in both and ectopic kidney in one patient) with the same MLPA kit used in our study [39]. The association between VSD and *GATA4* deletion was also reported in one case from a multicentric cohort [38]. Two cases of 8p23.1 duplication, including the *GATA4* gene, were found in a Hungarian cohort of children with CHDs, both members of the same family but with a different phenotypic appearance—one diagnosed with coarctation of the aorta and facial dysmorphism while the other presenting with patent foramen ovale and motor delay [31].

Part of the NK-2 homeodomain-containing family, the transcription factor *NKX2-5* is expressed in the first heart field, as well as the second heart field, indicating a central role in the regulation of the developing heart [51]. Studies have shown a specific involvement in the development of the outflow tract and the right ventricle while also regulating the formation of the conduction system [52,53,54]. Approximately 2–4% of all CHDs can be associated with SNPs in the *NKX2-5* gene [17], particularly linked with septal defects (ASDs and VSDs) or outflow tract defects (tetralogy of Fallot, double-outlet right ventricle, and transposition of the great arteries) [55,56]. CNVs involving this gene were also reported; one patient carrying a triplication of the 5q35.3 region, including the *NKX2-5* gene, was found through MLPA screening of a CHD multicentric cohort. The patient presented with ASD, pulmonary stenosis, as well as extracardiac findings (umbilical and inguinal hernias, hypospadias, brachydactyly, microcephaly, and psychomotor delay) [38].

*TBX5* is a member of the T-box family, and it is a transcription factor known to play an important role in regulating cardiomyocyte identity [14]. It is also involved in the septation of the heart, both for the atria and the ventricles [57], and activates the expression patterns that specify and regulate the development of the ventricular conduction system [58]. Haploinsufficiency of *TBX5* is the cause of Holt–Oram syndrome (HOS), an autosomal dominant disorder associated with congenital heart defects (mostly septal defects, the atrial septal defect being the most frequent), conduction abnormalities, and limb anomalies (bilateral and asymmetric radial ray defects) [59]. Although none of the studies that used MLPA as a CNV screening method for CHD patients found subjects with the involvement of the *TBX5* gene, both deletions and duplications involving this gene were reported in a number of patients with an atypical HOS phenotype through more complex genetic testing, especially array Comparative Genomic Hybridization [60,61,62,63].

To our knowledge, this is the first study to investigate the burden of CNVs in the 22q11.2 region and *GATA4*, *TBX5*, *NKX2-5*, *BMP4*, and *CRELD1* genes in a Romanian population of children with CHDs. The major limitation of our study was the relatively small number of patients enrolled in the cohort, which can explain the absence of less common CNVs, such as those present in the *GATA4*, *TBX5*, *NKX2-5*, *BMP4*, and *CRELD1* genes, in our patients.

In conclusion, our study evaluated the presence of CNVs in particular regions previously related with congenital heart defects (22q11.2 and the *GATA4*, *TBX5*, *NKX2-5*, *BMP4*, and *CRELD1* genes) by using a MLPA technique and determined the presence of 22q11.2 deletion in two cases, both from the syndromic group. No CNVs were reported in the isolated group. Our data support the diagnostic strategy that proposes MLPA as a screening test for patients with syndromic CHDs and suggest the inclusion of the 22q11.2 region in the panels used for screening. Further research is needed in order to evaluate the involvement of CNVs in the *GATA4*, *TBX5*, *NKX2-5*, *BMP4*, and *CRELD1* genes in the development of congenital heart defects, particularly in the Romanian population.

## Figures and Tables

**Figure 1 genes-15-00207-f001:**
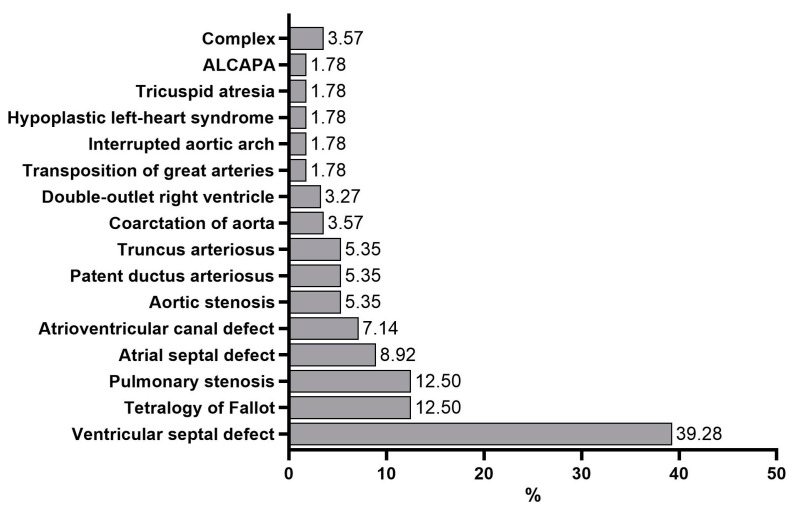
Distribution of the types of congenital heart defects in the studied cohort. (ALCAPA, anomalous left coronary artery from the pulmonary artery).

**Figure 2 genes-15-00207-f002:**
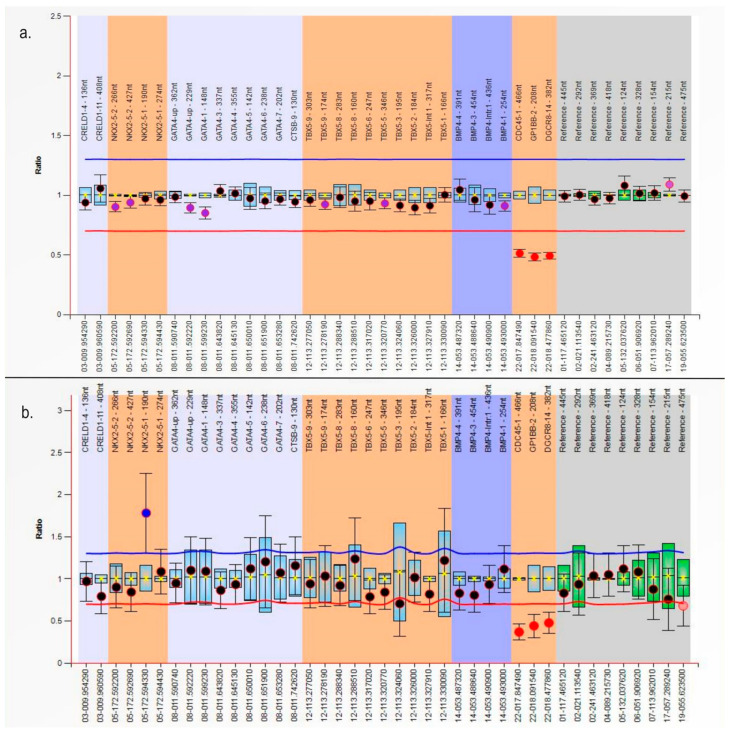
Multiplex ligation-dependent probe amplification (MLPA) analysis showing the subjects with 22q11.2 heterozygous deletion: (**a**) Case number 1; (**b**) Case number 2.

**Figure 3 genes-15-00207-f003:**
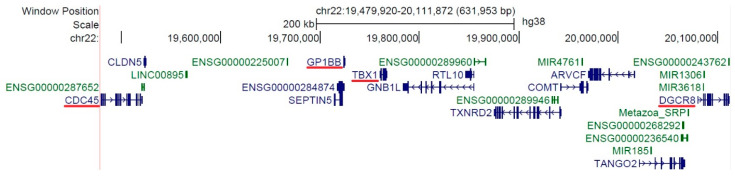
The genetic architecture of the deleted DNA fragment found in the 22q11.2 region from our patients; blue—coding genes, green—non-coding genes, and red line—genes of interest (generated using http://genome.ucsc.edu (accessed on 27 January 2024) [30]).

**Table 1 genes-15-00207-t001:** Demographical, clinical, and echocardiographic characteristics of the studied cohort.

Characteristic	Syndromic (28 Subjects)	Isolated (28 Subjects)	*p*-Value
Age ^1^ (months), median [Q1–Q3]	28 [5–90.25]	46 [8.5–122.3]	0.2584
Female gender, *n*	14/28	18/28	0.4182
Urban, *n*	16/28	14/28	0.7891
Consanguinity, *n*	1/28	0/28	NA
Family history of CHD, *n*	7/28	8/28	>0.9999
Gestational age (weeks), median [Q1–Q3]	38 [37–40]	38 [37–40]	0.9796
Birth weight Z-score, median [Q1–Q3]	−1.12 [−1.54–−0.34]	−0.43 [−1.41–0.33]	0.0866
Birth length Z-score, median [Q1–Q3]	−0.65 [−1.86–0.12]	0.41 [−0.65–1.57]	0.0237
SGA ^2^, *n*	3/28	3/28	>0.9999
Weight1 Z-score, median [Q1–Q3]	−2.18 [−3.6–−1.23]	−0.97 [−2.11–0.35]	0.0138
Height1 Z-score, median [Q1–Q3]	−1.25 [−2.4–−0.65]	−0.13 [−0.89–0.63]	0.0014
Short stature ^3^, *n*	6/28	3/28	0.4688
Type of CHD, *n*			
Ventricular septal defect	8/28	14/28	0.1707
Tetralogy of Fallot	5/28	2/28	0.4216
Pulmonary stenosis	4/28	3/28	>0.9999
Atrial septal defect	2/28	3/28	>0.9999
Atrioventricular canal defect	2/28	2/28	>0.9999
Aortic stenosis	1/28	2/28	>0.9999
Patent ductus arteriosus	0/28	3/28	NA
Truncus arteriosus	3/28	0/28	NA
Coarctation of aorta	0/28	2/28	NA
Double-outlet right ventricle	1/28	1/28	>0.9999
Transposition of great arteries	0/28	1/28	NA
Interrupted aortic arch	1/28	0/28	NA
Hypoplastic left-heart syndrome	1/28	0/28	NA
Tricuspid atresia	1/28	0/28	NA
ALCAPA	1/28	0/28	NA
Complex	1/28	1/28	NA

ALCAPA, anomalous left coronary artery from the pulmonary artery; CHD, congenital heart defect; SGA, small for gestational age. ^1^ At the time of blood sampling. ^2^ Birth weight with a Z-score < −2. ^3^ Actual height with a Z-score < −2.

**Table 2 genes-15-00207-t002:** Summary of the studies evaluating the MLPA detection of CNVs in patients with congenital heart defects.

Reference	Population	CNV	Frequency	Congenital Heart Defects
Mutlu et al., 2018 [35]	Türkiye	Deletion 22q11.2	3/45	ASD, ASD + VSD, VSD
Liu et al., 2015 [36]	China	Deletion 22q11.2	3/117	TOF
		Duplication 22q11.2	2/117	Endocardial cushion defect, PS + VSD
Nagy et al., 2019 [31]	Hungary	Deletion 22q11.2	2/49	TOF
		Duplication *GATA4* (8p23.1)	2/49	CoA, Patent foramen ovale
Li et al., 2018 [37]	China	Deletion 22q11.2	5/167	ASD + VSD, TOF, VSD
Sorensen et al., 2012 [38]	Denmark, Sweeden, France	Deletion *GATA4* (8p23.1)	1/402	VSD
		Deletion 22q11.2	2/402	PA + VSD, TOF
		Duplication 22q11.2	2/402	CoA, PS
		Triplication *NKX2-5* (5q35.3)	1/402	ASD + PS
		Complex (deletion *GATA4* + duplication 22q11.2)	1/402	AVSD + hypoplastic right ventricle
Floriani et al., 2020 [39]	Brazil	Deletion *GATA4* (8p23.1)	2/207	ASD + PLSVC + VSD, ASD + PLSVC + PS + VSD
		Deletion 22q11.2	4/207	ASD, TOF, VSD
		Duplication 22q11.2	1/207	Subvalvular aortic ring
Aguayo-Gomez et al., 2015 [40]	Mexico	No CNV found	0/52	-
Guida et al., 2010 [41]	Italy	No CNV found	0/161	-
Our study	Romania	Deletion 22q11.2	2/56	Interrupted aortic arch type B + VSD, VSD

ASD, atrial septal defect; AVSD, atrioventricular septal defect; CoA, coarctation of the aorta; CNV, copy number variation; PA, pulmonary atresia; PLSVC, persistence of left superior vena cava; PS, pulmonary stenosis; TOF, tetralogy of Fallot; VSD, ventricular septal defect.

## Data Availability

The data that support the findings of this study are available from the corresponding author upon reasonable request.

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
