# Peer review of "A Pilot Study of Multiplex Ligation-Dependent Probe Amplification Evaluation of Copy Number Variations in Romanian Children with Congenital Heart Defects"

_genes, 2024, doi:10.3390/genes15020207_

Round 1

Reviewer 1 Report

Comments and Suggestions for Authors

Dr Alexandru Cristian Bolunduț and collaborators investigated the involvement of Copy Number Variant in few genes encoding cardiac enriched transcription factors   known to cause congenital heart diseases (CHD) including the 22q11 locus including Tbx1 responsible for the DiGeorge syndrome. To such an aim, they used the Multiplex ligation-dependent probe amplification (MLPA)

They carried out their study using a cohort of 56 CHD patients. They found two patients with CNV in the 22q11 locus. They compared their findings with the literature.

The study is well designed and confirms the MLPA technique as an efficient approach to detect CNV.

 The data confirm previously published data as to the implication of CNV in a minority of CHD ( table 2 ) at least as  far as a few genes are investigated.

 The limitation of the study if indeed the very few genes investigated and the limited number of CHD patients. This could explain the low percentage of CNV in CHD.  Have the authors thought to look at other cardiac genes playing an early role in heart formation (Bves, Tcea3, Fhod3…) as well as genes encoding chromatin modifiers ?

Do the authors have the opportunity to expand their CHD cohort? This could provide them with more power in their investigation

Have the authors checks CNV in 22q11 in halthy patients ? this is a important control tp add 

Author Response

Thank you very much for your review and suggestions.

This study is part of a doctoral research, so being a pilot study and due to some financial constraints, we limited ourselves to this panel of genes. Also, we chose to perform a cohort study and include only patients with congenital heart defects as a more suitable research methodology in this case, due to the expected low frequency of these CNVs in healthy population. To our knowledge, this is the first study evaluating the burden of these CNVs in a Romanian population of congenital heart defects. Investigating also the healthy population and including a control group would be suitable direction that we will consider for further research in more ample projects, as well as expanding the panel of genes, because there are a great number of genes playing different roles in heart development.

Reviewer 2 Report

Comments and Suggestions for Authors

This study investigates the role of copy number variations (CNVs) in congenital heart defects (CHD), focusing on the 22q11.2 region and specific genes (GATA4, TBX5, NKX2-5, BMP4, CRELD1). In a cohort of 56 patients, half syndromic CHD cases, ventricular septal defect (VSD) is the most common. Using multiplex ligation-dependent probe amplification (MLPA) analysis, two 22q11.2 deletions are identified in syndromic cases. No CNVs in the specified genes are found. The study suggests MLPA as an initial genetic test for syndromic CHD, advocating for the inclusion of the 22q11.2 region in screening panels.

Comments and suggestions:

1.     Could the authors provide the sequences of used probes as supplemental materials?

2.     There are about 70 genes (including some miRNAs) locating the 22q11.2 region (PMID: 30380194); discussions should be included on which genes of these 70 possibly contribute to CHD in the study.

3.     Some more discussion should be added in terms of why mutations on GATA4, TBX5, and NKX2-5 genes usually cause CHD.

Author Response

Thank you very much for your suggestions. We included the sequences of used probes in a supplemental material and addressed the suggested discussions in the revised version of the manuscript.

Round 2

Reviewer 1 Report

Comments and Suggestions for Authors

no additional comment